# Allergy to Polyethilenglicole of Anti-SARS CoV2 Vaccine Recipient: A Case Report of Young Adult Recipient and the Management of Future Exposure to SARS-CoV2

**DOI:** 10.3390/vaccines9050412

**Published:** 2021-04-21

**Authors:** Vincenzo Restivo, Giuseppina Candore, Maria Barrale, Ester Caravello, Giorgio Graziano, Rosa Onida, Maurizio Raineri, Salvatore Tiralongo, Ignazio Brusca

**Affiliations:** 1Department of Health Promotion, Mother and Child Care, Internal Medicine and Medical Specialties, University of Palermo, 90127 Palermo, Italy; caravelloester@hotmail.com (E.C.); giorgio.graziano@gmail.com (G.G.); salvatore.tiralongo@unipa.it (S.T.); 2Department of Biomedicine, Neurosciences and Advanced Diagnostics, University of Palermo, 90127 Palermo, Italy; giuseppina.candore@unipa.it; 3Laboratory of Clinical Pathology, “Buccheri La Ferla” Hospital, 90127 Palermo, Italy; marbarrale@gmail.com (M.B.); rossellaonida@gmail.com (R.O.); brusca.ignazio@fbfpa.it (I.B.); 4Department of Biopathology and Medical Biotechnologies, University of Palermo, 90127 Palermo, Italy; maurizio.raineri@unipa.it

**Keywords:** SARS CoV2, vaccine, PEG, adverse reaction, m-RNA vaccine, anaphylaxis, contraindication, basophil activation test

## Abstract

The main contraindication to the anti-SARS CoV2 vaccine is an anaphylactic reaction to a vaccine component. The need to vaccinate allergic people who are at higher risk can be of public health interest and this report shows a case of an allergic reaction to PEG of a HCW who had received the first dose of anti-SARS CoV2 vaccine. For 5 h after the administration of the vaccine, she had the appearance of erythematous spots on the face and neck, and a feeling of a slurred mouth and hoarseness. In order to treat the event, she was administered 8 mg intravenous dexamethasone, 1 vial intravenous chlorphenamine maleate, 250 mL intravenous 0.9% NaCl, and conventional oxygen therapy (2 L/min) with complete resolution of the suspected adverse drug reaction. According to the contraindication to the cutaneous test for this patient, BAT was used for further investigations. The patient who suffered the adverse reaction to the COVID-19 vaccine and other five allergic patients who did not report any adverse reaction after the vaccination were tested. There was a significant activation of the vaccine-reactive patient’s basophils with 14.79 CD203chigh% at the concentration of 0.2 mg/mL, while other patients were negative. People who have a confirmed reaction to a vaccine component should undergo further investigation to discover other possible cross-reactions and select the right vaccine to immunize them.

## 1. Introduction

The eradication of SARS CoV2 will depend not only on the possibility of ensuring a wide distribution of effective vaccines but also on the population’s compliance with the vaccination itself. The main contraindication to anti-SARS CoV2 vaccine is an anaphylactic reaction to a vaccine component. Overall, anaphylactic reactions to vaccines are rare, accounting for about 1.31 cases per million doses [1]. Allergic reactions to vaccines are mostly directed to components such as stabilizers, adjuvants, preservatives, and residual contaminants from the production process. Knowledge of all the components of the formulation is useful in identifying the culprit allergen [2]. Although the anaphylactic reactions reported with the anti-SARS CoV2 vaccine were rarely observed in the various phases of the trial and in the early stages of vaccination campaigns, they seem to be more frequent and they have aroused concerns in the population [3].

As of February 2021, 10 vaccines have been authorized by at least one national regulatory authority for public use: two RNA vaccines, four conventional inactivated vaccines, three viral vector vaccines, and one peptide vaccine [4]. Pfizer-BioNTech recently introduced its anti-SARS CoV2 vaccine in the UK, US, and other countries. This is a vaccine based on messenger RNA (mRNA) (tozinameran, BNT-162b2) and the vaccine contains several excipients and lipids, one of which is based on PEG-2000. This is currently the only excipient of the vaccine with a recognized allergenic potential [5].

Since the approval of the mRNA-based vaccines, several case series have been published from the United Kingdom and the United States regarding individuals who have experienced severe general reactions in the setting of anti-SARS CoV2 vaccination. Even if episodes of anaphylaxis were not observed in clinical studies for either vaccine, there have been reports in the United States and other countries of anaphylactic reactions that occurred after administration during the mass vaccination campaigns. Based on US data from 1.8 million doses of the administered Pfizer-BioNTech vaccine, 175 cases of severe allergic reaction have been reported to the Vaccine Adverse Events Reporting System (VAERS), 21 of which were found to be compatible with anaphylaxis. Moreover, of these, 17 cases occurred in people with a history of allergic reactions, and 70% of the reactions occurred within 15 min of vaccine administration [3]. Furthermore, the Moderna vaccine administration reported 10 cases of possible anaphylaxis (2.5 cases of anaphylaxis per million doses administered) [6]. In particular, Pfizer-BioNTech has an anaphylaxis rate four times higher than the Moderna vaccine. Regarding the Oxford-AstraZeneca COVID-19 vaccine, the UK Medicines and Healthcare products Regulatory Agency (MHRA) had 234 reports of anaphylaxis from approximately 11.7 million vaccinations as of 7 March 2021 [7]. However, the vaccine components do not differ significantly and the reason for this observed increased reaction rate is unknown. This rate of anaphylaxis is significantly higher than was previously reported for one million doses of other vaccines, but is still quite rare (11.1 cases per million vaccine doses of the Pfizer-BioNTech vaccine).

Healthcare workers (HCWs) are more exposed to the SARS CoV2 virus and they are entitled as the first category for vaccination due to their working activities [8]. Although allergic reactions to the PEG contained in the anti-SARS CoV2 vaccine have been little treated in the scientific literature, the need to vaccinate allergic people who are at higher risk can be of public health interest. This report shows the case of an allergic reaction to PEG in an HCW who had received the first dose of the anti-SARS CoV2 vaccine.

## 2. Case Presentation

A young Caucasian woman, a 30-year-old radiologist resident physician, had clear signs of an allergic reaction at the first administration of the Pfizer-BioNTech vaccine.

She was a polyallergic subject with a reported urticaria-angioedema episode in 2009 and multiple other immediate cutaneous reaction elicited by chocolate, honey, some cosmetics, and detergents. The first reaction arose a few hours after the ingestion of a meal containing shrimps. Accurate anamnesis revealed a suspected LTP sensitization. The hygiene products that elicited an immediate reaction (urticaria), in the patient can contain PEG. Moreover, the subject had been administered, about 15 years ago, three doses of a vaccine containing polysorbate without any reaction.

The patients had not ingested food for many hours before the vaccine administration and, due to the positive anamnesis for allergic reaction, she was advised to take the following premedication: two tablets of 25 mg prednisone 14 h before the administration of the vaccine, two tablets of 25 mg prednisone 7 h before, and two tablets of 25 mg prednisone and a vial i.m. of chlorphenamine maleate (10 mg) 1 h before vaccination. On the other hand, she reported self-administering prednisone (25 mg, one tablet) 14 h before, one tablet 7 h before and one tablet with an i.m. vial (10 mg) of chlorphenamine maleate 1 h before the administration of the vaccine.

After the administration of the vaccine, she was observed for 30 min without any clinically significant alterations. For 5 h after the administration of the vaccine (injected in the left arm), she had the appearance of erythematous spots on the face and neck and the feeling of a slurred mouth and hoarseness. In order to treat the event, she went to the vaccination service. She was administered, by physicians of the Rapid Response System (RRS), 8 mg intravenous dexamethasone, one vial intravenous chlorphenamine maleate, 250 mL intravenous 0.9% NaCl, and conventional oxygen therapy (2 L/min), with complete resolution of the suspected adverse drug reaction.

Lab tests were performed to analyze the inflammatory, chemical, and physical parameters. The tests showed a total IgE value of 214 KU/L (normal range: 0–100), a 0% eosinophil rate (normal range: 0–8%), a WBC value of 7.89 × 10^3^/Ul (normal range: 4–11), a 93% neutrophil rate (normal range: 40–74%), a 0.5% monocyte rate (normal range: 3–11%), a 6% lymphocyte rate (normal range: 20–48), and a 0.4% basophile rate (normal range: 0–1.5%). Furthermore, the serum tryptase assay showed a value lower (2.03 μg/L) than the cut-off (10 μg/L). Another serum tryptase assay conducted in basal conditions almost a month later revealed a value of 3.55 μg/L.

The presence and titer of serum antinuclear antibody (ANA) were assayed on HEp-2 slides at a starting serum dilution of 1:80 and positive samples were further diluted to a final titer of 1:1280. The serum samples which were positive for ANA at 1:1280 dilution were further processed for detection of antibodies against double-stranded DNA (anti-dsDNA) and extractable nuclear antigen (ENA), by *Crithidia luciliae* IFA and an immunoblotting assay, respectively, as recommended by ANA Reflex. Furthermore, the C3 fraction of the complement had a value (96 mg/dL) that was at the lower limits of the normal range (90–180).

The patient was also counselled to have a visit by the allergist team that examined the patient 2 weeks after the resolution of clinical symptoms. The patient showed an old dosage of specific IgE with positivity to house dust mites, peach, and other fruits. The tests were performed by the CLIA method on an Immulite 2000 instrument (Siemens Heathcare AG, Erlangen, Germany). Component resolved diagnosis was not performed, nor was the tryptase dosage. She suffered of dermatographism that would not permit the execution of a skin prick test. Beyond the dermatographism, the patient did not suffer from aquagenic urticaria. Specific IgE (Immunocap Thermo Scientific, Uppsala, Sweden), perfomed 1 month after the allergic reaction, confirmed the positivity of house dust mites and Pru p3, while shrimp tropomyosin resulted negative.

### Basophil Activation Test

Flow cytometric quantification of “in vitro basophil activation” after allergen incubation has been demonstrated as a reliable tool to evaluate the IgE-dependent allergen-specific and non-IgE-mediated reaction responses in allergic patients [9]. The basophil activation test (BAT) has been recently approved by the World Allergy Organization for the diagnosis of allergic diseases [10]. Due to its diagnostic characteristics, BAT has been specifically proposed for the diagnosis of vaccine allergies [11]. According to the contraindication to the cutaneous test for this patient, BAT was used for the diagnosis. This test was already available as a PEG 4000 allergen that was prepared by BÜHLMANN Laboratories AG, Schönenbuch, Swiss, specifically designed for BAT using CD63 as a basophil degranulation marker. The concentration in the vial of PEG 4000 was 4 mg/mL. The BAT protocol was adapted using CD203c as basophil activation marker (Allergenicity kit, Beckman Coulter, Brea, California). The CD203c activation requires a lower concentration of allergen than CD63 because the molecule marked by CD203c is associated with low-dose events of chemotaxis and it is not a basophil degranulation marker [12]. These differences were useful for choosing the PEG concentrations to be tested.

The patient who suffered the adverse reaction to the COVID-19 vaccine and another five allergic patients who did not report any adverse reaction after the vaccination were tested (females; medium age: 31.2 years; range: 27–35).

The BAT was performed using the manufacturer’s instructions. Briefly, samples of whole blood in EDTA were incubated for 15 min after the addition of sufficient amounts of calcium solution to override the chelating capacity of EDTA, with PEG 4000 at the following concentrations: 4 mg/mL, 0.8 mg/mL, 0.4 mg/mL, 0.2 mg/mL, 0.1 mg/mL, and 0.08 mg/mL. Anti-IgE antibody at 4 μg/mL and PBS were used as a control. PC7-conjugated anti-CD3, FITC-conjugated anti-CRTH2, and PE-conjugated anti-CD203c antibodies were added during the incubation. The samples were analyzed on a DxFlex Flow Cytometer (Beckman Coulter, Brea, California). The basophils were detected and counted on the basis of forward side scatter characteristics and the expression of negative CD3 and positive CRTH2. Upregulation of CD203c on cells was established using a threshold defined by the fluorescence on PBS stimulated cells (negative control). At least 500 basophils were analyzed at each assay. Data are expressed as CD203chigh%. Figure 1 shows the dose–response curve of all patients. There was a significant activation of the vaccine-reactive patient’s basophils with 14.79 CD203chigh% with the concentration of 0.2 mg/mL, while other patients were negative.

## 3. Discussion

This report describes the adverse reaction to PEG after the first dose of anti-SARS CoV2 vaccine administered to an HCW. The Pfizer-BioNTech vaccine was administered to an HCW whoa never confirmed an allergy to PEG or other components of the vaccine. Furthermore, the Astrazeneca vaccine, suggested as an alternative to the Pfizer-BioNTech vaccine, was approved by the EMA only on 29 January, 3 weeks after the patient’s vaccination. Nowadays, the FDA advises that the vaccine should be contraindicated in patients with a severe allergic reaction to the first dose of vaccine or with known hypersensitivity to any ingredient/component of the vaccine. However, it is possible to consider resuming vaccination using other types of vaccine. Especially in case of a severe anaphylactic reaction, it is preferable not to administer the second dose of the same vaccine and consider the possibility of restarting the vaccination from scratch using other types of vaccine (with different excipients, after carrying out skin tests) [13].

The premedication scheme for allergic patients with a history of episodes of anaphylaxis used for the patient before vaccination is similar to that used for other procedures as a radiodiagnostic test with a contrast medium. This therapy, acting by the modulation of gene expression, is generally not significant for 20–30 min. Moreover, when administered intravenously, the HCWs engaged with it must have specific skills in diagnosis and treatment of anaphylactic reactions [14]. Furthermore, the indications provided in premedication schemes must be considered without any solid evidence, and the guidelines suggest premedication in a few circumstances with very low evidence [15]. The main explanation may be that no adequate randomized clinical trials have been conducted due to ethical reasons. Moreover, the effectiveness in the patient with vaccine adverse reaction may have been reduced by the half dosage of prednisone, probably invalidating the efficacy of the premedication.

The immediate systemic reactions reported after the anti-SARS CoV2 vaccine can be caused by several pathological mechanisms. In addition to IgE-mediated reactions, non-IgE-mediated mechanisms via the MAS-related G protein-coupled receptor X2 (MRGPRX2) or complement-dependent activation pathways (complement activation-related pseudoallergy = CARPA) may play a role which cannot necessarily be detected by prick test.

PEG is discussed as the trigger of the reactions. No reactions to PEG in vaccines have been reported, but PEG has not been a commonly used excipient in vaccines until now [15]. It is bound to a liposomal matrix, which is a nanoparticle that coats the viral mRNA of the COVID-19 vaccine. The PEG with a molecular weight of 2000 (PEG2000) has the role of preventing premature degradation of the nanoparticles by the mononuclear phagocytosis system, and also as a solubilizer during the transition of the particles into the intracellular cytosol due to its hygroscopic properties, and as an adjuvant due to its immunogenic potential [5].

Considering the widespread PEG use in numerous products for daily use (cosmetics, drugs, laxatives, tablets), hypersensitivity to PEG is very rare. Various allergic manifestations have been described as late reactions such as allergic contact dermatitis, and also contact urticaria and anaphylaxis as immediate reactions. However, there are no studies on the prevalence of immediate hypersensitivity to PEGs, nor is its real incidence known. Only one study analyzed 10 patients diagnosed with PEG allergy [16]. Eight patients had experienced at least one anaphylactic reaction requiring adrenaline treatment, caused by antibiotic/analgesic tablets, depot-steroids, antacids, and laxatives. Seven patients reported repeated reactions before diagnosis (median: 3; range: 2–6). None of the patients experienced severe allergic reactions after the diagnosis. The median Likert score of the impact on daily life before diagnosis was 7 in comparison with 4 after diagnosis [16].

When a non-anaphylactic allergic reaction (urticaria and/or angioedema and/or generalized itching and/or mild respiratory symptoms) occurs after the first dose of a vaccine, the allergic consultation provides for skin allergometry. If the skin tests are positive for PEG, it is also recommended to perform tests with polysorbate 20 and 80 to evaluate the described cross-reactivity, and the possibility of using future COVID-19 vaccines and another vaccine must be considered [17,18]. The polysorbates are also widely used as excipients of liquid and solid formulations of drugs and are capable of inducing both immunologically and non-immunologically mediated anaphylactic reactions [19]. It should be noted that polysorbate 80, which is commonly found in influenza vaccines and is also present in non-mRNA-based anti-SARS CoV2 vaccines (less than 100 μg/dose), is potentially cross-reactive to PEGs and may also cause anaphylactic reactions [20,21]. In the United Kingdom, it has been recommended that patients with PEG allergy can receive the Astra-Zeneca vaccine under 30 min of observation, whereas in the United States, the CDC most recently classified a previous reaction to polysorbate 80 as a contraindication to an mRNA-based vaccine [21]. Unfortunately, the Astra-Zeneca vaccine was not available in Italy when the HCW in this report was vaccinated.

According to the European EAACI guidelines, the level of tryptase is essential to exclude the presence of mastocytosis and to characterize the anaphylactic events that may occur during vaccination [22]. Furthermore, it is useful to perform the test in the acute phase (i.e., 30 min after the event) with its repetition in basal conditions. An increase in tryptase is considered significant if the acute tryptase results are higher than the basal tryptase level × 1.2 (+2). Although it is not possible to exclude the diagnosis of an allergic reaction, the tryptase value detected in relation to the patient’s medical history did not suggest a mast cell disease.

A value of 1:1250 of ANA was also found, but its association with a lack of ENA was unable to diagnose an autoimmune state. The absence of rheumatologic disease reduces the reported accuracy of the tests among the 3–5% of healthy people that are ANA-positive [23]. Furthermore, the reduced value of the C3 fraction of the complement can be indicative of inflammatory and infectious diseases (rheumatoid arthritis, SLE, subacute bacterial endocarditis) [24].

## 4. Conclusions

Overall, allergic reactions to vaccines remain exceedingly rare and the allergic patients deserve access to the same publicly recommended vaccinations as non-allergic patients unless the risks associated with the vaccination outweigh the gains. At this stage, it is important that events such as these do not lead to misinterpretations and detract from global implementation of the vaccine. People that have a confirmed reaction to a vaccine component should undergo further investigation to discover other possible cross-reactions and select the right vaccine to immunize people with an allergic history. The BAT in the patient’s reported history identified the reactivity towards PEG. However, it is necessary to collect further data to evaluate the sensitivity and specificity performances of BAT towards PEG.

## Figures and Tables

**Figure 1 vaccines-09-00412-f001:**
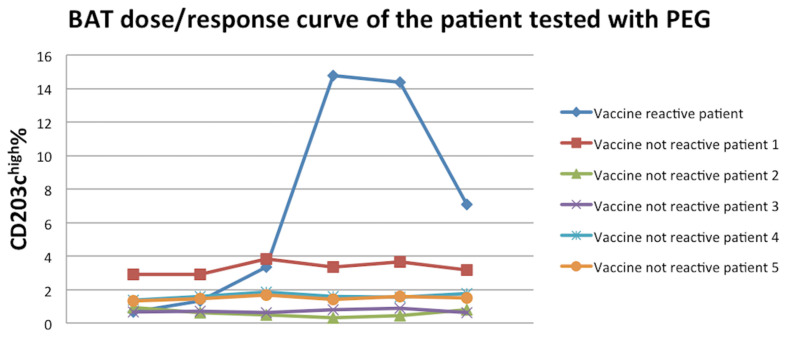
BAT dose–response curve of the patients tested with PEG.

## Data Availability

The data presented in this study are available on request from the corresponding author.

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
