# Peer review of "Allergy to Polyethilenglicole of Anti-SARS CoV2 Vaccine Recipient: A Case Report of Young Adult Recipient and the Management of Future Exposure to SARS-CoV2"

_vaccines, 2021, doi:10.3390/vaccines9050412_

Round 1
Reviewer 1 Report
Restivo et al report on the case of a health care worker showing an allergic reaction to one of the mRNA vaccines against SARS-CoV-2.
Given the persistence of the COVID-19 pandemic and the urgency of vaccinations against the new coronavirus and its variants, reports of adverse events and approaches to better understand their causes are of very high relevance. Thus the report is very timely.
However the manuscript is missing the figure and requires significant English language editing.
Author Response
Dear reviewer 1, I added the figure and revised the English language.
Reviewer 2 Report
The manuscript by Vincenzo et al, on “Allergy to polyethilenglicole of anti-SARS CoV2 vaccine recipi-2 ent: a case report of young adult recipient and the management 3 of future exposure to SARS-CoV2.” The study is well designed and the data support the conclusion made and the methods used are well documented. The reviewer has very few minor suggestions.
- Introduction part should be descriptive, include addition information.
- Include sufficient references.
- Make a graphical representation for improve quality of article.
Author Response
- Introduction part should be descriptive, include addition information.
1. A. The introduction was enriched with other information as following “Even if episodes of anaphylaxis were not observed in clinical studies for either vaccine, there have been reports in the United States and other country of anaphylactic reactions that occurred after the administration during the mass vaccination campaigns. Based on US data from 1.8 million doses of the administered Pfizer-BioNTech vaccine, 175 cases of severe allergic reaction have been reported to the Vaccine Adverse Events Reporting System (VAERS) whose twenty-one were found to be compatible with anaphylaxis. Moreover, of these, 17 cases occurred in people with a history of allergic reactions, and 70% of the reactions occurred within 15 minutes of vaccine administration [6]. Furthermore, the Moderna vaccine administration reported 10 cases of possible anaphylaxis (2.5 cases of anaphylaxis per million doses administered) [7]. In particular, Pfizer-BioNTech has an anaphylaxis rate four times higher than Moderna vaccine. Regarding the Oxford-AstraZeneca COVID-19 vaccine the UK Medicines and Healthcare products Regulatory Agency (MHRA) had 234 reports of anaphylaxis on approximately 11.7 million vaccinations as of March 7, 2021 [8]. However, the vaccine components do not differ significantly and the reason for this observed increased reaction rate is unknown. This rate of anaphylaxis is significantly higher than previously reported for one million vaccination doses of other vaccines, but is still quite rare (11.1 cases per million vaccine doses of Pfizer-BioNTech vaccine).”
2.Q.Include sufficient references.
A. Additional references was added related both to introduction and discussion section.
3. Q. Make a graphical representation for improve quality of article.
A. The graphical representation was added to the manuscript.
Reviewer 3 Report
The authors present a case study of an allergic reaction of a HCW who received the Pfizer-Biontech vaccine. The manuscript flags up the importance of investigating such patients and the suitability of the Basophil activation test (BAT). This is of interest but there are aspects of the study which need further discussion with appropriate references.
- Line 42. When the authors talk about the anti-SARS CoV2 vaccines they should be more specific and make clear that they only mean the Pfizer-Biontech vaccine. Also, what do they mean by more frequent? More frequent compared to what? The authors need to compare frequency of anaphylactic reactions to other vaccines Covid 19 vaccines currently used.
- Line 54. Again, which vaccine or vaccines are the authors referring to that gives 11.1 cases per million.
- Line 70. There is no explanation of why a patient who may have a possible reaction to PEG would have been given the Pfizer-Biontach vaccine. In line 150, the authors do in fact state that the FDA advise that the vaccine (again need to make clear which vaccine or vaccines) is contraindicated in patients with known hypersensitivity to any ingredient/component of the vaccine.
- Line 73. The steroid treatments given before vaccination would certainly reduce the individual’s response to the vaccine so it is quite surprising that this course of action was taken. There is no discussion around this aspect in the manuscript. If available in Italy’ the Astra Zeneca vaccine might have been a better choice for this individual. In line 193 the authors do say that in the UK that would have been the course of action. More discussion around vaccine availability in relation to this patient would therefore be important.
Author Response
- Q. Line 42. When the authors talk about the anti-SARS CoV2 vaccines they should be more specific and make clear that they only mean the Pfizer-Biontech vaccine. Also, what do they mean by more frequent? More frequent compared to what? The authors need to compare frequency of anaphylactic reactions to other vaccines Covid 19 vaccines currently used.
A. Dear reviewer 3, I thank you first of all for your advices. I believe they will be very constructive and could make this article more impactful. We added the following to the introduction section Even if episodes of anaphylaxis were not observed in clinical studies for either vaccine, there have been reports in the United States and other country of anaphylactic reactions that occurred after the administration during the mass vaccination campaigns. Based on US data from 1.8 million doses of the administered Pfizer-BioNTech vaccine, 175 cases of severe allergic reaction have been reported to the Vaccine Adverse Events Reporting System (VAERS) whose twenty-one were found to be compatible with anaphylaxis. Moreover, of these, 17 cases occurred in people with a history of allergic reactions, and 70% of the reactions occurred within 15 minutes of vaccine administration [6]. Furthermore, the Moderna vaccine administration reported 10 cases of possible anaphylaxis (2.5 cases of anaphylaxis per million doses administered) [7]. In particular, Pfizer-BioNTech has an anaphylaxis rate four times higher than Moderna vaccine. Regarding the Oxford-AstraZeneca COVID-19 vaccine the UK Medicines and Healthcare products Regulatory Agency (MHRA) had 234 reports of anaphylaxis on approximately 11.7 million vaccinations as of March 7, 2021 [8]. However, the vaccine components do not differ significantly and the reason for this observed increased reaction rate is unknown. This rate of anaphylaxis is significantly higher than previously reported for one million vaccination doses of other vaccines, but is still quite rare (11.1 cases per million vaccine doses of Pfizer-BioNTech vaccine).
2. Q. Line 54. Again, which vaccine or vaccines are the authors referring to that gives 11.1 cases per million.
A. As you can see in the modified text, it was referred to the Pfizer Biontech vaccine.
3. Q. Line 70. There is no explanation of why a patient who may have a possible reaction to PEG would have been given the Pfizer-Biontech vaccine. In line 150, the authors do in fact state that the FDA advise that the vaccine (again need to make clear which vaccine or vaccines) is contraindicated in patients with known hypersensitivity to any ingredient/component of the vaccine.
A. Although the patient reported some hypersensitivity, she was never diagnosed an allergy to PEG or other components of the Pfizer-Biontech vaccine before its administration. Furthermore, the Astrazeneca vaccine, suggested as alternatives to the Pfizer-Biontech vaccine, was approved by the EMA only on January 29th, three weeks after the patient's vaccination. This explanations were reported in the discussion section.
4. Q. Line 73. The steroid treatments given before vaccination would certainly reduce the individual’s response to the vaccine so it is quite surprising that this course of action was taken. There is no discussion around this aspect in the manuscript. If available in Italy’ the Astra Zeneca vaccine might have been a better choice for this individual. In line 193 the authors do say that in the UK that would have been the course of action. More discussion around vaccine availability in relation to this patient would therefore be important.
A. The indications and effectiveness of the steroid treatment was reported in the discussion section. “The premedication scheme for allergic patients with a history of episodes of anaphylaxis used for the patient before vaccination is similar to which used for other procedures as radio diagnostic test with contrast medium. This therapy acting by the modulation of gene expression is generally not significant for 20-30 minutes, also when administered intravenously and the HCWs engaged with it must have specific skills in diagnosis and treatment of anaphylactic reactions [15]. Furthermore, the indications provided on pre-medication schemes must be considered without any solid evidence and the guidelines suggest the premedication in few circumstances with a very low evidence [16]. The main explanation can be that no adequate randomized clinical trials have been conducted due to ethical reasons. Moreover, the effectiveness in the patient with vaccine adverse reaction can be reduced by the half dosage of prednisone, probably invalidating the efficacy of the premedication”.
Furthermore the explanation of the use of Pfizer-Biontech vaccine for this patient was reported in the discussion section. as reported in the previous comment.
Round 2
Reviewer 1 Report
The manuscript now gives a lot more of the required background information and rationale for the approach with regard to the patient. There are some parts where the English needs to be significantly improved. e.g the new information starting at line 183.